Global marine biodiversity in the context of achieving the Aichi Targets: ways forward and addressing data gaps

Saeedi Hanieh hanieh.saeedi@senckenberg.de hanieh.saeedi@gmail.com 1 2 3
Reimer James Davis 4
Brandt Miriam I. 5
Dumais Philippe-Olivier 6
Jażdżewska Anna Maria 7
Jeffery Nicholas W. 8
Thielen Peter M. 9
Costello Mark John 10
1 Senckenberg Research Institute and Natural History Museum , Frankfurt am Main , Germany
2 FB 15 Biological Sciences Institute for Ecology, Diversity and Evolution Biologicum, Goethe University of Frankfurt , Frankfurt am Main , Germany
3 Senckenberg Research Institute and Natural History Museum, OBIS Data Manager, Deep-sea Node , Frankfurt am Main , Germany
4 Marine Invertebrate Systematics & Ecology Laboratory, Faculty of Science, University of the Ryukyus, Nishihara , Okinawa , Japan
5 IMARBEC, fremer, IRD, CNRS, Univ. Montpellier , Sète , France
6 Benthic Ecology Laboratory, Biology Department, Université Laval , Québec , Canada
7 Laboratory of Polar Biology and Oceanobiology, Department of Invertebrate Zoology and Hydrobiology, Faculty of Biology and Environmental Protection, University of Lodz , Lodz , Poland
8 Fisheries and Oceans Canada, Bedford Institute of Oceanography , Dartmouth , Nova Scotia , Canada
9 Research and Exploratory Development Department, Johns Hopkins Applied Physics Laboratory , Laurel , MD , United States of America
10 Institute of Marine Science, University of Auckland , Auckland , New Zealand
Marquet Pablo
Electronic publication date: 2019 Oct 29
Publication date: 2019
Volume: 7
Electronic Location ID: e7221
Received 2019 Feb 5; Accepted 2019 May 31
Copyright: ©2019 Saeedi et al.
Copyright year: 2019
Copyright holder: Saeedi et al.
License: This is an open access article distributed under the terms of the Creative Commons Attribution License, which permits unrestricted use, distribution, reproduction and adaptation in any medium and for any purpose provided that it is properly attributed. For attribution, the original author(s), title, publication source (PeerJ) and either DOI or URL of the article must be cited.
License URL: https://creativecommons.org/licenses/by/4.0/

Keywords: Aichi targets, Marine biodiversity, Prediction, Discovery, Biodiversity tools and pipelines, Biogeography, Data standard, Stewardship and dissemination, Stewardship, Data standards, Dissemination, Tools and pipelines, Marine biodiversity

Funding: NSERC Canadian Healthy Oceans Network and its Partners: Department of Fisheries and Oceans Canada INREST (representing the Port of Sept-Îles and City of Sept-Îles) This research is sponsored by the NSERC Canadian Healthy Oceans Network and its Partners: Department of Fisheries and Oceans Canada and INREST (representing the Port of Sept-Îles and City of Sept-Îles). The funders had no role in study design, data collection and analysis, decision to publish, or preparation of the manuscript.

==============================
In 2010, the Conference of the Parties of the Convention on Biological Diversity agreed on the Strategic Plan for Biodiversity 2011–2020 in Aichi Prefecture, Japan. As this plan approaches its end, we discussed whether marine biodiversity and prediction studies were nearing the Aichi Targets during the 4th World Conference on Marine Biodiversity held in Montreal, Canada in June 2018. This article summarises the outcome of a five-day group discussion on how global marine biodiversity studies should be focused further to better understand the patterns of biodiversity. We discussed and reviewed seven fundamental biodiversity priorities related to nine Aichi Targets focusing on global biodiversity discovery and predictions to improve and enhance biodiversity data standards (quantity and quality), tools and techniques, spatial and temporal scale framing, and stewardship and dissemination. We discuss how identifying biodiversity knowledge gaps and promoting efforts have and will reduce such gaps, including via the use of new databases, tools and technology, and how these resources could be improved in the future. The group recognised significant progress toward Target 19 in relation to scientific knowledge, but negligible progress with regard to Targets 6 to 13 which aimed to safeguard and reduce human impacts on biodiversity.

Introduction

The ‘Strategic Plan for Biodiversity 2011–2020’ of the Convention on Biological Diversity was agreed during the 10th Conference of the Parties, held from 18 to 29 October 2010, in Aichi Prefecture, Japan. The strategic plan included five main “Strategic Goals” that were divided into 20 targets. Each ‘Aichi Target’ was designed to better understand and predict biodiversity dynamics, such as how biological diversity underpins ecosystem function, and how the provision of ecosystem services is essential for human well-being. Meeting the Aichi Targets would ultimately benefit local livelihoods and economic development, and is essential for biodiversity maintenance and poverty reduction (Shepherd et al., 2016; Tittensor et al., 2010). In this paper, we report on the findings of a working group that discussed how the targets related to marine biodiversity were being achieved.

Survey Methodology

The 4th World Conference on Marine Biodiversity (WCMB 2018) in Montreal organised a working group to review and evaluate how the Aichi Targets have been met by the global community. In particular, the review group was asked to focus on Target 19 regarding scientific knowledge about biodiversity as this was the area of most expertise of the participants. To identify and reduce biodiversity knowledge gaps we examined how marine biodiversity discoveries and their predictions need to be redirected to better understand and predict how marine biodiversity will change within the next 10 years. Here, we identify seven important priorities for this topic to support the Aichi Targets (Table 1). These foci arose iteratively from discussion between the group members and other groups at the conference. The priorities address issues of (1) data standards, (2) education in data management, (3) taxonomic expertise, (4) genetic tools, (5) international collaboration, (6) identifying knowledge gaps and understanding biogeography, and (7) the need to reduce human pressures on marine biodiversity.

Table 1 The examined Aichi Targets.

The Aichi Targets examined in this study and their relationship to the scientific priorities identified by participants of the 2018 World Conference on Marine Biodiversity. The Targets were grouped into Strategic Goals by the Convention on Biological Diversity. * negligible progress, ** notable progress, *** good progress.

Aichi target	Scientific priority	Progress	
	Developing, improving and enhancing biodiversity data standards, exchange, and analytical tools, via standardized techniques.	***	
	Educational activities to increase data mobilisation by taxonomists, data users, and/or wider audiences.	***	
Strategic Goal E: Enhance implementation through participatory planning, knowledge management and capacity building	Promoting synergy of biodiversity research efforts via increased collaboration at all levels.	***	
Target 19
By 2020, knowledge, the science base and technologies relating to biodiversity, its values, functioning, status and trends, and the consequences of its loss, are improved, widely shared and transferred, and applied.	Utilization and promotion of taxonomic expertise and species identification tools to better recognize and catalogue biodiversity.	**	
	Improvement and standardization of genetic, genomic, and other “omics” tools to aid in discovery, assessment, description, and cataloging of biodiversity.	**	
	Identifying biodiversity and biogeographic knowledge gaps and promoting efforts to reduce such gaps.	***	
Strategic Goals A: Address the underlying causes of biodiversity loss by mainstreaming biodiversity across government and society [1] B: Reduce the direct pressures on biodiversity and promote sustainable use, Targets 6 to 10 [2] and C: To improve the status of biodiversity by safeguarding ecosystems, species and genetic diversity, Targets 11–13 [3]	Control of anthropogenic pressures on vulnerable ecosystems impacted by climate change or ocean acidification to maintain their integrity and functioning.	*	
Notes.

[1] Target 3: By 2020, at the latest, incentives, including subsidies, harmful to biodiversity are eliminated, phased out or reformed in order to minimize or avoid negative impacts, and positive incentives for the conservation and sustainable use of biodiversity are developed and applied, consistent and in harmony with the Convention and other relevant international obligations, taking into account national socio economic conditions. Target 4: By 2020, at the latest, Governments, business and stakeholders at all levels have taken steps to achieve or have implemented plans for sustainable production and consumption and have kept the impacts of use of natural resources well within safe ecological limits.

[2] Target 6: By 2020 all fish and invertebrate stocks and aquatic plants are managed and harvested sustainably, legally and applying ecosystem-based approaches, so that overfishing is avoided, recovery plans and measures are in place for all depleted species, fisheries have no significant adverse impacts on threatened species and vulnerable ecosystems and the impacts of fisheries on stocks, species and ecosystems are within safe ecological limits. Target 7: By 2020 areas under agriculture, aquaculture and forestry are managed sustainably, ensuring conservation of biodiversity. Target 8: By 2020, pollution, including from excess nutrients, has been brought to levels that are not detrimental to ecosystem function and biodiversity. Target 9: By 2020, invasive alien species and pathways are identified and prioritized, priority species are controlled or eradicated, and measures are in place to manage pathways to prevent their introduction and establishment. Target 10: By 2015, the multiple anthropogenic pressures on coral reefs, and other vulnerable ecosystems impacted by climate change or ocean acidification are minimized, so as to maintain their integrity and functioning.

[3] Target 11: By 2020, at least 17 per cent of terrestrial and inland water, and 10 per cent of coastal and marine areas, especially areas of particular importance for biodiversity and ecosystem services, are conserved through effectively and equitably managed, ecologically representative and well-connected systems of protected areas and other effective area-based conservation measures and integrated into the wider landscapes and seascapes. Target 12: By 2020 the extinction of known threatened species has been prevented and their conservation status, particularly of those most in decline, has been improved and sustained. Target 13: By 2020, the genetic diversity of cultivated plants and farmed and domesticated animals and of wild relatives, including other socio-economically as well as culturally valuable species, is maintained, and strategies have been developed and implemented for minimizing genetic erosion and safeguarding their genetic diversity.

Results and Discussion

Reviewed priorities

Developing, improving and enhancing biodiversity data standards, exchange, and analytical tools, via standardized techniques

Recent marine biodiversity discoveries have been greatly enhanced by standardised open-access taxonomic and biogeographic data repositories such as the World Register of Marine Species (WoRMS) and the http://www.iobis.org/ (OBIS) (Ahyong et al., 2018; Costello et al., 2013a). Large-scale development of marine biodiversity data standards and exchange started when the Census of Marine Life (2000–2010) established OBIS (O’Dor et al., 2012).

Biodiversity data standards, such as “Darwin Core”, a data schema that provides stable terms and vocabularies for universal sharing of biodiversity data, and management techniques have been improved to ensure that published data have high quality. For example, there are now available taxonomic and geographic data management tools and R packages, e.g., rOBIS (Provoost, 2018) (Fig. 1). The taxon match tool in WoRMS can automatically check the correct spelling, authority, classification and validity of species names uploaded to a webpage (Costello et al., 2013a). A marine gazetteer matches place names to geographic coordinates and polygon (shapefiles) of localities (Claus et al., 2014). The standardization and open storage of metadata, taxonomic, genetic, and geographic data also allows for greater stewardship by stakeholders, enhanced public awareness and education, and importantly, the ability to easily share data among institutions (Fig. 1).

Figure 1 Biodiversity data processing.

Biodiversity data processing using novel analytical standardized techniques and technologies.

Developing biodiversity data standards and data exchange protocols enables both data users and providers to benefit from the high-quality data that later allow for more reliable and precise biodiversity analyses. The expansion of the OBIS data schema to include additional information associated with sampling events, including sampling methods and environmental data, is a significant recent advance (De Pooter et al., 2017). The open-access publication of thousands of data sets integrated into OBIS has enabled major advances in our understanding of global patterns of biodiversity. For example, several studies have utilized open-access marine species distribution records to discover and confirm large-scale biodiversity patterns. These findings include observations that global species richness is bimodal with latitude, and that species richness decreases with depth (Chaudhary, Saeedi & Castello, 2016; Chaudhary, Saeedi & Costello, 2017; Costello & Chaudhary, 2017; Saeedi, Basher & Costello, 2016; Saeedi & Costello, 2019a; Saeedi & Costello, 2019b; Saeedi, Dennis & Costello, 2017; Saeedi et al., in press; Hillebrand, 2004). We conclude that there has been noticeable progress towards achieving the potential of Aichi Target 19 through establishing global databases and taxonomic resources. These are leading to better quality control, data management efficiencies, and new insights into our understanding of biodiversity. However, these benefits, as well as contributions to these resources, are not realised in all countries.

Educational activities to increase data mobilisation by taxonomists, data users, and/or wider audiences

Many scientists and the general public around the world are unaware of the presence and advantages of large-scale open-access databases. Even if scientists are familiar with these facilities, data preparation and submission can be complex for contributors unfamiliar with data publication protocols. These issues are more pronounced for scientists in developing countries or non-native English speakers. As a result of these perceived and real data publication obstacles, significant biodiversity and biogeography knowledge remains in personal databases and non-digital archives. These logistical hurdles and data ownership perceptions frequently stand in the way of data publication. To expose researchers to these resources, initiatives like OceanTeacher Global Academy (OTGA) (https://classroom.oceanteacher.org) or Ocean School (https://oceanschool.nfb.ca/) provide a valuable educational platform that aids sustainable development. OceanTeacher is part of the http://www.iode.org/ Programme of the http://ioc.unesco.org/. It has trained nearly 2,000 students from 120 countries since 2005 (https://classroom.oceanteacher.org/mod/page/view.php?id=2033). OTGA has hosted OBIS training workshops to train data providers on how to prepare, standardize, and submit their data to OBIS, where their data is cited and safely secured. Organisations such as OTGA need to be financially supported by governments in order to actively educate and train data keepers and encourage them to share their data with the global community. However, as the scientific community is relatively small and financial sources for training are limited, the future of large-scale biodiversity studies is likely to rely on the well-designed application of citizen science in addition to technological advances (Stuart-Smith et al., 2017; Thiel et al., 2014).

Promoting synergy of biodiversity research efforts via increased collaboration at all levels

In order to predict and discover biodiversity on a global scale, collaborative approaches among institutions and nations are necessary. Guralnick, Hill & Lane (2007) proposed a framework to use online databases and tools to improve and standardize geographic data, and to validate and highlight taxonomic data and misidentifications. They also suggested that a global infrastructure for web-based tools would enhance the quality of visualizing and standardizing raw biodiversity data and lead to a higher degree of collaboration and accessibility of knowledge (Guralnick, Hill & Lane, 2007).

The decade long Census of Marine Life was the largest global collaboration amongst marine biologists (O’Dor et al., 2012). Its legacy continues in OBIS with regard to data publication, but also continued international collaboration amongst polar, deep-sea and other researchers. The International Association for Biological Oceanography (IABO) is the organisation officially responsible for coordinating the marine biodiversity community (Costello et al., 2015a). It runs the MARINE-B email list with over 1,000 subscribers, and holds a World Conference on Marine Biodiversity every three years. Many others, often more specialist-focused, conferences also serve to bring marine biodiversity researchers together. These serve to make introductions and help build collaborative relationships among researchers. However, most research funding is for topics of national rather than international importance. For example, not every country will have specialist expertise in every taxonomic group. Thus, sharing of taxonomic expertise can alleviate funding deficits, allow the transfer of knowledge, and lead to international partnerships.

The number of marine species formally described each year has never been greater, and aside from naming these species, more work is required to understand their life histories and ecology, biogeography, and evolution (Appeltans et al., 2012). Costello, Vanhoorne & Appeltans (2015) recommended the use of collaborative online databases, taxonomic effort improved and increased through communication, easier access to specimens, engagement of non-specialists, and international collaboration (Costello, Vanhoorne & Appeltans, 2015). Further, Costello et al. (2013b) advocated abandoning “data-sharing” and instead suggested requiring data publication within a journal or to online infrastructures such as OBIS, WoRMS, and/or the Global Biodiversity Information Facility (GBIF) (Costello et al., 2013b). A fundamental aspect of Aichi Target 19, namely discovering the full extent of biodiversity in the world’s oceans, is not possible without international collaboration.

Utilization and promotion of taxonomic expertise and species identification tools to better recognize and catalogue biodiversity

“Good” taxonomy is an absolute necessity for biodiversity recognition and management (Thomson et al., 2018). It is very important to pass on the knowledge of experienced taxonomists to others. In this regard, field-specific training workshops can be of great importance. As an example, two ‘IceAGE (Icelandic marine Animals: Genetics and Ecology) amphipod identification workshops’ were recently held, consisting of two weeks of work of a group of taxonomists accompanied by students. This resulted in the identification of more than 20,000 amphipod specimens, and the publication of seven research papers dealing with the taxonomy, diversity and ecology of this group around Iceland (Brix et al., 2018).

Another problem in recognizing biodiversity is the complexity of access to information and images that can help end-users in identifying the organisms collected in their samples. To solve this requires comprehensive online identification guides to all species on Earth (Costello, May & Stork, 2013b). It is surprising that despite the numerous online digital initiatives regarding biodiversity resources for this most practical need (how to identify species) lack leadership and remain scarce. Targeted funding to support such resources is urgently needed to help the wider community identify species quickly and accurately, including species that may be invasive or pathogenic. Despite great publicity and interest, publically available DNA libraries, such as the Barcode of Life (Ratnasingham & Hebert, 2007), are still far from having complete coverage of the Tree of Life. Moreover, DNA is often only useful if the species has already been formally described, specimens from which DNA has been sampled have been correctly identified, and the DNA sequence(s) published in an open access database. Other tools that can help in species identification for conservation and resource management include interactive keys. Unfortunately, there are only such keys for a few marine taxa, and these mostly concern higher taxonomic levels (family-level or higher) (Dallwitz, 1997; Nimbs, 2017).

Along with Aichi Target 19, the exchange of knowledge between experienced scientists and young researchers as well as the use of different identification tools (e.g., interactive keys, barcode databases) will help in biodiversity recognition. These targets need to be prioritised by governments and funding agencies as they are fundamental to quality assurance in other areas of biology, ecology, and resource management.

Improvement and standardization of genetic, genomic, and other “omics” tools to aid in discovery, assessment, description, and cataloging of biodiversity

Countless studies have used genomic tools to study marine biodiversity, connectivity, and functional diversity (Carvalho et al., 2010). While data sharing, standardized sampling, metadata collection, and sequencing protocols still require significant standardization, the use of repositories such as GenBank, Dryad, and Sequence Read Archive have made published data much more accessible. We expect this trend to continue as new tools, such as the Genomic Observatories Metadata database (GeOMe), streamline sequencing data and metadata submission (Deck et al., 2017).

The classification and phylogeny of eukaryotic organisms benefits from the use of genetic markers that help delineate species in combination with morphological, ecological, and/or physiological information (Hebert, Ratnasingham & De Waard, 2003; Palumbi, 2003). Currently, no uniform threshold value has been established for species delineation, and there is no single “universal” DNA barcode that captures all eukaryotic life. Even within a single animal order there can be large differences in this value between families (Tempestini, Rysgaard & Dufresne, 2018). Despite these issues, DNA barcoding and next-generation metabarcoding can reveal genetic diversity and are useful tools for the description and cataloguing of biodiversity.

Modern high-throughput sequencing (HTS) technologies have advanced DNA barcoding methods by producing millions of individual sequences per analyzed sample, enabling DNA metabarcoding from environmental DNA (eDNA) and complex community mixtures. Community and environmental metabarcoding are both useful tools to discover cryptic (undetected until genetic analyses) diversity in the marine realm, and assess ocean biodiversity in a non-invasive and high-throughput manner (Goodwin et al., 2017).

In order to obtain robust and reproducible metabarcoding results, critical methodological aspects remain to be improved (Dickie et al., 2018; Goldberg et al., 2016; Nichols et al., 2018). Studies are needed to address the effects of alternative protocols on sampling, molecular, and bioinformatic processing level in order to develop standardized and reliable techniques for applying these new methods. Furthermore, because large amounts of marine organisms have not been genetically characterized, integrative approaches should be supported in order to fill database gaps. The continued improvement and standardization of genetic, genomic, and other “-omics” tools (e.g., proteomics, transcriptomics) will continue to be valuable components in the discovery of new marine prokaryotes and eukaryotes, as well as in monitoring biodiversity, thereby contributing to Aichi Target 19.

Identifying biodiversity and biogeographic knowledge gaps and promoting efforts to reduce such gaps

Deep-sea ecosystems include about 65% of the world’s surface but are far less studied and sampled than shallower depths (Costello et al., 2018; Costello, Cheung & Hauwere, 2010). Although deep-sea studies have increased rapidly in recent decades, there are large gaps in global sampling coverage, for example in the Indian and Pacific Oceans, and major efforts are needed to continue to be directed into offshore research (Saeedi et al., 2019; Barroso et al., 2018). The distribution and diversity of deep-sea fauna thus still remains poorly-known due to the size and remoteness of deep-sea ecosystems. For example, recent studies have shown that the global latitudinal marine species richness gradient follows a bi-modal pattern related to temperature and habitat availability (Chaudhary, Saeedi & Castello, 2016; Chaudhary, Saeedi & Costello, 2017; Saeedi, Basher & Costello, 2016; Saeedi & Costello, 2019a; Saeedi & Costello, 2019b; Saeedi, Dennis & Costello, 2017). This finding is supported by the fossil record, which shows reduced species richness at the equator in warm periods (Kiessling & Aberhan, 2007; Yasuhara et al., 2012). As such, the peaks in this bimodal distribution may become further separated under future climate change and ocean warming. However, there is still no consensus about this bimodal pattern in the deep sea, where food supply may be more important than temperature in defining species distribution. For example, Woolley et al. (2016) examined 165,000 distribution records of Ophiuroidea and revealed that biogeographic patterns in species richness in the deep sea are associated with chemical energy and proximity to slope habitats. However, these patterns require investigation in other taxa, from micro- to mega-fauna, epifauna and infauna (Woolley et al., 2016; Clarke & Gaston, 2006).

One issue in studying present and future global deep-sea biodiversity patterns are the few publicly available distribution records and environmental data. Deep-sea expeditions began over 100 years ago, but distribution data are often still retained in inaccessible archives, sometimes in local languages of that country, and are not publically available to the global community. Additionally, acquisition of data describing the distributions of deep-sea species is often limited by prohibitive costs and logistical difficulties in surveying the deep ocean. Environmental suitability modeling has thus become a cost-effective tool for identifying potential locations of deep-sea species, particularly for areas that have never been explored (Assis et al., 2018; Basher & Costello, 2016; Danovaro et al., 2017; Serrano et al., 2017).

Understanding how abiotic drivers influence species distributions can contribute to filling spatial gaps of biodiversity hotspots and endangered areas (McHenry, Steneck & Brady, 2017; Saeedi, Basher & Costello, 2016; Saeedi & Costello, 2019a; Saeedi & Costello, 2019b; Saeedi, Dennis & Costello, 2017). Since some of these drivers can be observed by satellite imagery, it is possible to model some community assemblages in difficult-to-access locations. The development of models of species richness and cumulative anthropogenic impact distributions could be useful for conservation purposes and/or other spatial planning applications (Selig et al., 2014). Costello et al. (2018) proposed that sampling of the oceans should be stratified in relation to environmental variability, with more variable environments receiving more sampling focus in space and time (Costello et al., 2018). Coupled with a recent objective (data driven) delineation of marine biodiversity into 30 biogeographic realms based on the endemicities of marine plants and animals (Costello et al., 2018; Costello, Cheung & Hauwere, 2010; Costello, Vanhoorne & Appeltans, 2015), this provides a framework for more representative sampling of the oceans. These realms include 18 continental-shelf and 12 offshore realms, including unique seas, such as the Baltic and Black seas, and subdivisions of the Indian, Atlantic, and Pacific oceans, and polar waters. These are broad scale patterns, and application of stratified sampling at local levels requires finer spatial resolution data; such as benthic habitat maps and observations of movements of threatened species to know where they occur.

Control of anthropogenic pressures on vulnerable ecosystems impacted by climate change or ocean acidification to maintain their integrity and functioning

Fisheries and marine mammal hunting have had large impacts on biodiversity at local scales for centuries, and at global scales for the past two centuries (Pauly et al., 2002). Clearly fishery management measures struggle to prevent overfishing, and trawling that destroys seabed habitats is widely permitted, while bycatch of seabirds, turtles and marine mammals is pushing some species to extinction (McCauley et al., 2015; Mittelbach et al., 2007). Progress in reducing bycatch is compromising reaching the achievement of Aichi Target 12 related to preventing species extinctions. A proven solution to reversing some negative trajectories are marine reserves (no-take MPAs) (Costello, 2014). However, about two-thirds of coastal countries lack even one marine reserve, and over 90% of MPAs allow fishing and thus prevent the recovery of biodiversity to natural conditions (Costello & Ballantine, 2015). This failure to conserve and help fisheries recover, despite the potential benefits of MPAs to nature, education, science (they act as controls for effects of fishing outside them), tourism, and fish populations defies what is best for society. With less than 3% of the ocean in reserves, there seems little hope that Aichi Target 11′s goal of 10% of the oceans being protected in MPAs by 2020 will be reached. In addition, there appears to be negligible progress towards more sustainable use of the oceans, as called for in Targets 4, 6 and 7. Target 3, the reduction of harmful subsidies, has also seen little progress and too many fisheries still receive indirect and/or direct subsidies from governments that enable further unsustainable overfishing.

Aichi Target 10 calls for reduced anthropogenic impacts on coral reefs. Coral reefs suffered global-scale bleaching events in 2015–2017, even within MPAs, resulting in massive damage to these ecosystems, including mass mortality of hermatypic corals and other zooxanthellate organisms (Hughes et al., 2017), and associated reduced ecosystem functioning (Hughes, Kerry & Simpson, 2018; Hughes et al., 2018). Additionally, such events have economic effects such as reduced tourism (Prideaux, Carmody & Pabel, 2017). Overall, the trajectory of coral reefs continues to be one of downward degradation in the face of increasing anthropogenic pressures, including climate change and continued exploitation (Heery et al., 2018).

Other anthropogenic impacts on marine biodiversity include excess nutrient input, oxygen depletion, and invasive species. Levels of these impacts are to be reduced and their management improved as part of Aichi Targets 8 and 9. Progress in management of introduced and invasive marine species has been made with the establishment of the World Register of Introduced Marine Species (WRiMS) (Ahyong et al., 2018). Because of the nature of invasive species, management of their information is most cost-effectively done at a global rather than local scale. The next steps should include access to species identification resources and a dynamic online reporting and early warning system.

Both global warming and ocean acidification are closely linked with the anthropogenic input of CO2 and other greenhouse gases into the atmosphere, and without controlling these issues, the future of coral reefs looks bleak IPCC2018. Minimizing anthropogenic impacts such as increased runoff from coastal development and reducing overfishing can help delay the degradation of coral reef ecosystems, but it is estimated more than half of all coral reefs now experience medium to high anthropogenic pressures (Halpern et al., 2008) and the extirpation of species from many coral reefs due to climate change is predicted (Molinos et al., 2016).

There are some success stories, such as Palau, which has passed stringent legislation protecting coral reef diversity, including the world’s first no-take zone for sharks (Vianna et al., 2012), stringent legal protection (Gouezo et al., 2017), and a visitor’s pledge and public awareness campaign (https://palaupledge.com/). Other regions or countries following the lead of these exemplars could help buy time for coral reef ecosystems. For instance, the Australian government implemented the Great Barrier Reef (GBR) Zoning Plan 2003 in 2004, which set aside one-third of the GBR as a no-take zone (McCook et al., 2010). This resulted in a significantly lower proportion of reefs being affected by Crown-of-thorns starfish outbreaks in no-take zones than in fished zones (McCook et al., 2010), but the trajectory of GBR coral reef ecosystems remains bleak due to warming-associated coral bleaching (Hughes, Kerry & Simpson, 2018; Hughes et al., 2018).

Conclusions

While there has been considerable progress in addressing many of the priorities of the Aichi Targets, including the development and application of biodiversity tools and higher standards, as well as increased educational activity and increasing standardization of genetic and genomic tools, progress towards sustainable use is very limited (Table 1). Of the seven priorities for marine biodiversity to achieve the Achi Targets we reviewed here, we judged six have seen some progress. However, other goals such as reducing anthropogenic stressors on vulnerable ecosystems have clearly not been met and seem certain to fall short of the 2020 Targets, as previously concluded (Tittensor et al., 2014). We recommend continued efforts regarding international cooperation in marine biodiversity informatics to increase efficiencies of data management and accessibility, paralleled by field observations stratified by environmental conditions and societal needs, synergised by online access to taxonomic information and species identification guides, and research into more cost-efficient methods for field sampling that minimise impacts on biodiversity (e.g., sensors, video, eDNA). This surveillance of marine biodiversity should occur in areas where human impacts, particularly fishing, occur, but also in fully-protected Marine Protected Areas where no fishing occurs (i.e., marine reserves). The lack of progress towards establishing the latter reference areas compromises our understanding of natural conditions and how to judge the sustainability of human activities in the ocean and their impacts on marine biodiversity.

We would like to thank David Beauchesne, Rémi M. Daigle, Jésica Goldsmit, Philippe Archambault, Anna Metaxas, and Paul Snelgrove for their thoughts and leadership of the mentorship program organizing committee and workshop facilitators. Special thanks also to Rémi M. Daigle and Jésica Goldsmit for reviewing the current paper. Comments from Martin Thiel and Naomi Kingston greatly improved an earlier version of this paper.

Additional Information and Declarations

Competing Interests

Author Contributions

Data Availability

James D. Reimer and Mark J. Costello are Academic Editors for PeerJ.

Hanieh Saeedi and Mark John Costello conceived and designed the experiments, prepared figures and/or tables, authored or reviewed drafts of the paper, approved the final draft, draft and revised some paragraphs of the current paper.

James Davis Reimer conceived and designed the experiments, authored or reviewed drafts of the paper, approved the final draft, draft and revised some paragraphs of the current paper.

Miriam I. Brandt, Philippe-Olivier Dumais, Anna Maria Jażdżewska, Nicholas W. Jeffery and Peter M. Thielen conceived and designed the experiments, draft and revised some paragraphs of the current paper.

The following information was supplied regarding data availability:

We do not have any raw data as this is a review article.

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
