# Peer review of "Global marine biodiversity in the context of achieving the Aichi Targets: ways forward and addressing data gaps"

_PeerJ, doi:10.7717/peerj.7221_

## Round 0.1 · original submission · Major Revisions

I concur with both reviewers in the importance of this manuscript but also in the need to make an effort to better support the achievement, or lack thereof, of the Aichi targets, especially for coastal marine environments.

·

Basic reporting

see below

Experimental design

see below

Validity of the findings

see below

Additional comments

Comments to ms by Saeedi et al. (see also annotated manuscript):

General evaluation: This manuscript explores the different strategic goals of the AICHI protocol for biodiversity discovery and prediction. In general, this can probably be a useful overview, offering some suggestions for future research and conservation strategies. However, during my reading I had the impression that the treatment of the different priorities/goals is a bit superficial and subjective – there might be options to overcome this by using some approach (see below) to make this a bit more objective (keeping in mind that there will always remain some subjectivity in this kind of forum pieces). Additionally, I think that it would be good to characterize this piece from the outset as a forum or opinion piece – possibly this could be highlighted in the title.

A general impression: from reading the different sections on the 8 research priorities, I had the feeling that the evaluation about achievements was that in many cases things were in poor or (maximally) fair state, but then in the final conclusion sections the authors say “we judge seven have seen fair to good progress”. Clearly, my impression after reading the sections maybe as subjective as the judgement by the authors. It would be really good if the authors made an attempt to make these evaluations of the 8 priorities/goals a bit more objective. I think they could define several categories (e.g. very poor, poor, fair, good, very good, excellent) and then for each of the 8 priorities/goals conclude each section by saying something like e.g. “Based on the above considerations and criteria, we evaluate that the achievement of this priority/goal is fair.” And then, in the final conclusion section, they could provide an overview table that summarizes all these judgements. I fully understand that a rigorous and objective evaluation would probably be a major task and would go beyond this overview, but I do think that there are some options to achieve a somewhat more objective approach. As expressed in this paragraph, there is some contrast between the reading and the concluding judgement, and an effort should be made to avoid this apparent contrast.

I would also recommend to place a bit more emphasis on coastal zones in areas of the world’s oceans that continue to be very poorly studied. In many parts of the world our knowledge of local (coastal) biodiversity is poor, and there are many obstacles to improve that situation (economical, political, taxonomic expertise, etc.) and that should be highlighted, especially given that most anthropogenic stressors severely impact the coastal zones. While the mentioning of the deep sea gaps and threats is certainly important, a better balance between deep sea examples and mentioning of poorly studied coastal zones would be good.

Also, I would be more cautious with painting MPAs as the panacea for conservation of marine biodiversity. There are certainly some success stories, but there are many other examples where MPAs actually cannot provide sufficient protection against external threats. The authors later acknowledge this when they talk about coral reefs, but some warning statement should also be included in the section where they emphasize the usefulness of MPAs.

Finally, I would encourage the authors to include at least a brief statement about the power of citizen science (see also comment in annotated manuscript). There are many examples how citizen scientists are contributing to a better knowledge of local marine biodiversity. Also, the power of this approach in engaging many members of the general public in the conservation of marine biodiversity could be mentioned.

In summary, the manuscript may be very timely, but it has ample room for improvements. Furthermore, it probably should be categorized from the outset as a forum or comments piece (in the title), to immediately distinguish it from a hard-core research paper. Incorporating my suggestions will require some time and effort, but it will make the paper much more powerful thereby ensuring that it will have a much stronger impact.

Finally, I hope that my comments will be helpful in improving the manuscript. In case of any questions concerning my comments, I invite the authors to contact me directly.

Sincerely, Martin Thiel, Facultad de Ciencias del Mar, Universidad Católica del Norte, Larrondo 1281, Coquimbo, Chile; email: [email protected]

ps: i just notice that i can only upload the PDF of the annotated manuscript to the PEERJ website - i would be happy to provide the annotated word document any time - just send me an email.

·

Basic reporting

Overall the abstract reflects the text of the paper well, but needs some careful editing. There is one issue with the framing of the problem, which implies a fundamental misunderstanding of the objectives behind the Strategic Plan for Biodiversity and the Aichi targets. These are global targets for action rather than simply analytical targets for studies. Some reframing of the abstract will correct this.

The paper has been written by multiple authors and could do with some careful editing for language and consistency.

Experimental design

Line 65 - This makes no sense as drafted. They are global targets, so need to be met by the global community. Reporting to the targets is carried by countries who each contribute through their national biodiversity strategies and plans. Scientists certainly play an important role and feed into that process by undertaking studies that contribute to reporting and developing indicators to inform the targets. But scientists can't 'meet' the targets per se.

Line 72 - The focal areas identified in the paper are indeed key areas that need further focus, but these underpin many more of the Aichi targets than set out in Table 1.

Also it is important to view the targets alongside the indicators used to measure them if the aim is to understand the status of achievement through a scientific lens. Some aspects may be achieved, but we simply don't have the data to show that, or as in the case of target 11 we are basing achievement on only a subset of the target!

Validity of the findings

Throughout the paper there is some confusion over the correct terminology for the 'Strategic Plan for Biodiversity 2011-2020', and also the purpose and ambition of this global plan. Suggest the addition of a co-author with expertise in this area.

The summary of the discussion topics is very interesting and a useful contribution to the scientific literature. However, I think the framing as a response to Aichi targets is unconvincing.

While stated as a paper that discusses the achievement of the Aichi targets, the link between the targets and measurable indicators is absent from the paper. Indeed to link form the focal areas to the targets feels like an afterthought rather than an integral part of the study design.

Additional comments

Overall there is useful content in this paper that should be published. However, it has been framed poorly as a response to delivery of the Aichi targets and in this is fails.

The paper is based on a discussion group at the 4th World Conference on Marine Biology. The group identified 8 focal areas for further discussion. The focal areas seem to have been arrived at as focal areas for future research and development areas. It is not clear how they linked to the Aichi targets during this discussion.

In the paper they review each of the focal areas, but the connection to the Aichi targets feel much like an afterthought and not part of the core study design. Indeed there is no mention of the indicators that are used to measure the Aichi targets, which would seem a critical area for the authors to review if they are looking at how best to measure and achieve the targets for the focal areas.

I would like to see the authors rethink the framing of the paper. Instead focus on how the reviewed priority areas could be better reflected in the Aichi targets. It would be very useful to use the results to propose new indicators and ways of measuring progress. This can then make a clear link to the drivers of positive change. This would be a very worthwhile contribution to the literature, and timely given that the targets will likely be revised in 2020 and there is a strong drive to make the post2020 targets more measurable.

---

## Round 0.2 · accepted · Accept

The revision addresses all the concerns raised by reviewers, so I commend the author for doing a great job and addressing all issues identified by reviewers. This is a timely and important contribution.